# Effect of Creatine and Glucose on Formation of Heterocyclic Amines in Grilled Chicken Breasts

**DOI:** 10.3390/foods8120616

**Published:** 2019-11-25

**Authors:** Monika Gibis, Myriam Loeffler

**Affiliations:** Department of Food Physics and Meat Science, Institute of Food Science and Biotechnology, University of Hohenheim, Garbenstrasse 21/25, 70599 Stuttgart, Germany; Myriam.Loeffler@uni-hohenheim.de

**Keywords:** heterocyclic amines, precursor, glucose, ribose, creatine, chicken, carcinogens

## Abstract

The occurrence of heterocyclic amines (HAs) in grilled chicken breasts was investigated. All samples contained HAs, including MeIQx, PhIP, and the β-carbolines harman and norharman. In particular, PhIP was found in concentrations of 1.5–9.1 ng/g, and MeIQx was detected at very low concentrations (n.d.–1.1 ng/g). The concentrations of two co-mutagenic β-carbolines, harman and norharman, ranged from 0.8 to 2.3 ng/g when the content of the precursor glucose was varied from 100 to 620 mg/kg. In contrast, the content of the precursor creatine in non-grilled chicken breasts varied by only 8.6%. A significant linear correlation existed between the molar concentration of PhIP and the molar ratio of creatine/glucose (*r* = 0.88, *p* < 0.001). We, thus, conclude that the formation of PhIP may be inhibited with increasing concentrations of glucose in chicken breast. Chicken patties coated with ribose or glucose-containing water in oil emulsions confirmed that both reducing sugars decreased PhIP formation with the preferred concentrations (sensory analysis) of 0.5–1% for ribose and 1% for glucose leading to a reduction of PhIP formation by 28–34% and 39%, respectively.

## 1. Introduction

The variety of forms of human nutrition and the associated content of consumed food carcinogens may be one factor for the observed variation in rates of cancer across the world. Based on sufficient evidence in humans that the intake of processed meat causes colorectal cancer [1,2], the International Agency for Research on Cancer (IARC) has classified the consumption of red meat as probably carcinogenic to humans (Group 2A), and of processed meat as carcinogenic to humans (Group 1). Nevertheless, there is insufficient evidence for the presence of some known carcinogenic compounds in meat and meat products.

Heterocyclic amines (HAs) were found in the crust of fried, broiled, grilled, or roasted meat and fish [3,4]. Some HAs are known to be mutagenic in the Ames test with *Salmonella* Typhimurium and were reported to be 100 times more mutagenic than Aflatoxin B1 [3]. Additionally, long-term animal studies on rodents or non-human primates (given dose: 10 to 20 mg IQ per kg body weight in the feed) showed that HAs are carcinogenic at multiple sites, including liver, prostate, lungs, breast, and the colorectum [5]. Several epidemiological studies have also shown that the frequent consumption of HAs may lead to an increased risk of cancer developing [2]. Furthermore, the detection of DNA adducts in human tissues in vivo, such as the mammary gland, colon, pancreas, and mucosa [6,7], suggest a genotoxic effect. The IARC has classified several HAs as possible or probable human carcinogens and has suggested reducing the daily intake of HAs. Additionally, the California Environmental Protection Agency defines the following NSRLs (no significant risk levels) for HAs (MeIQx, 0.41; MelQ, 0.46; Glu-P-1, 0.1; Glu-P-2 and IQ, 0.5; AαC, 2; MeAαC, 0.6; Trp-P-1, 0.03, and Trp-P-2 0.2 µg/day); for PhIP, no NSRL exists, the agency plans to define NSRL for PhIP in the next 5 years [8]. These NSRLs represent the daily intake level calculated to result in a cancer risk of one excess case of cancer in 100,000 persons exposed over a lifetime of 70 years [8]. 

HAs such as imidazoquinoline, -quinoxaline, and -pyridine (IQ-type) are usually formed from creatine/creatinine, amino acids, and carbohydrates in reactions that are part of the Maillard Reaction. In addition to the precursors of the raw material, physical parameters such as heating time and temperature, heat transfer, and general preparation methods [3], have an important impact on the formation of HAs. The experts of IARC assume that, during high temperature cooking, contaminates such as HAs or polyaromatic hydrocarbons are formed due to the exposure to hot surfaces or flames [1]. Recent studies, however, have not provided enough data for the IARC working group to make a clear statement on how meat should best be prepared and cooked to reduce cancer risk [1]. The IARC only classified red meat as probably carcinogenic to humans (Group 2A) [1] but did not classify white meat, such as poultry accordingly, even though chicken also contains HAs after cooking. The HAs occurring most often in meat are PhIP, MeIQx, 4,8-DiMeIQx, and, rarely, IQ, MeIQ, and AαC [9]. Co-mutagenic β-carbolines (harman and norharman) may also be present. The concentrations of MeIQx in normally cooked meat usually range between 1 and 5 ng/g, but concentrations up to 23 ng/g have been reported [10]. The PhIP concentrations described in most studies are between 1 and 70 ng/g meat [4,11], with the highest concentrations being found in chicken [12,13]. In a comparison of diverse cooking methods of chicken breasts, and compared to other HAs, pan-frying resulted in very high concentrations of PhIP (up to 18.3 ng/g) [14]. In the first reaction step of the formation of PhIP, phenylacetaldehyde is formed from phenylalanine via the Strecker degradation, and this then reacts in an aldol reaction with creatinine to form an intermediate product. PhIP arises from this latter intermediate in a condensation reaction [15]. In this reaction, creatinine is the cyclization product of creatine [16]. The longer the heating time, the more creatine is converted to creatinine [17]. In addition, the role of glucose depends on the available concentration, with small amounts of glucose increasing the formation of PhIP. In contrast, an inhibitory effect of glucose can be observed with the presence of excessive molar quantities compared with those of the other precursors [18]. 

The objective of this study was to examine the occurrence of the precursors glucose and creatine/creatinine and HAs in grilled chicken breasts. Additionally, the effect of coating with glucose- and ribose-containing water in oil emulsions (marinades) was studied. As a hypothesis, it was postulated that the addition of glucose or ribose to the emulsions results in a reduction of the concentration of PhIP in coated chicken breast patties during grilling. 

## 2. Materials and Methods

### 2.1. Materials and Chemicals

The left and the right breast chicken muscles (*M. pectoralis superficialis*) without skin from 21 birds were chosen for the investigations. Chicken breasts and refined sunflower oil were obtained from a local retailer (Mega, Stuttgart, Germany). The breasts were obtained from female poulards (commercial German hybrid breed) with an age of approximately 125 days and a sales weight of 1400 g. The chickens had already reached sexual maturity but were not ready to lay eggs. The visual fat and sinews were removed from the breasts prior to grilling. 

Norharman, harman, and caffeine (internal standard) were obtained from Sigma-Aldrich (Taufkirchen, Germany). IQ, MeIQ, IQx, MeIQx, 7.8-DiMeIQx, 4.8-DiMeIQx, PhIP, AαC, MeAαC, Trp-P-1, Trp-P-2, Glu-P-1, and Glu-P-2 were purchased from Toronto Research Chemicals (North York, Canada). Acetonitrile (gradient grade), methanol (gradient grade), aqueous ammonia (25%), toluene, and ethyl acetate (gradient grade), were purchased from Carl Roth GmbH & Co (Karlsruhe, Germany), and hydrochloric acid, ammonium acetate, sodium or potassium hydroxide, orthophosphoric acid, perchloric acid, and triethylamine from VWR International (Darmstadt, Germany). All the chemicals were analytical grade. Extraction blank cartridges Isolute^®^ and Isolute^®^ HM-N (diatomaceous earth) were obtained from Separtis GmbH (Grenzach-Wyhlen, Germany). Bond Elut^®^ C18, 100 mg and 500 mg Bond Elut^®^ PRS (Varian, Palo Alto, CA), and filter discs (type 0967, 11 mm ID) were obtained from Schleicher & Schuell GmbH (Dassel, Germany). Creatine kinase, lactate dehydrogenase, pyruvate kinase, adenosine-5′-triphosphate (ATP), and nicotinamide adenine dinucleotide (NADH) were obtained from Roche Diagnostics GmbH (Mannheim, Germany), and the determination kits of glucose, creatine, and creatinine were from r-biopharm AG, (Darmstadt, Germany).

### 2.2. Preparation and Heating of Chicken Breasts

Chicken breasts (poulard breasts) from several birds (*n* = 21) were cut into equal-sized rectangular slabs (about 10 cm long × 14 cm wide × 1.3 cm thick). An electric contact grill (Model Nevada, Neumärker GmbH & Co. KG, Hemer, Germany) was heated to a plate temperature of 220 °C. The chicken breasts were lightly brushed with refined sunflower oil to avoid adherence to the aluminum foil. Each chicken breast was covered with two pieces of aluminum foil and then grilled for about 3 min to give a core temperature of 72 °C and a surface temperature ≤198 °C. Surface and core temperatures were monitored during the grilling process using a temperature data logger (Almemo^®^ 8990-8, Ahlborn, Holzkirchen, Germany). Each of the two chicken breasts from one bird was grilled between two griddle plates of the electric contact grill, and each chicken breast was separately homogenized afterward. In these grill processes, the heat transfer is influenced by direct conduction and, to a lesser extent, by convection. 

### 2.3. Preparation and Coating of Chicken Patties

Ten chicken breasts were coarsely minced using a meat grinder (Stephan Machinery GmbH, Germany) with a plate with 3 mm holes. After the addition of salt (12 g/kg) and pepper (1 g/kg) and mixing, 80 g ± 1 g of the minced material was formed into patties using a special patty mold (12 mm deep × 85 mm diameter). Afterward, the patties were coated (marinated) with water in oil (W/O) emulsions containing 0.5, 1, 2.5, 5, 10, or 20 wt% of glucose and ribose, respectively. The W/O emulsions were manufactured by homogenizing 67.5 g sunflower oil, 0.5 g emulsifier (citric acid esters of mono- and diglycerides of fatty acids, E472c, BASF, Illertissen, Germany) and 32 mL water [19]. The respective sugars were subsequently dissolved in the emulsion to give the required concentration, which represents the percentage by weight of sugar in the total W/O solution. Each patty was then coated on each side with approximately 1 g of the respective sugar emulsion (approximately 1.25–5 mg/g patty) (*n* = 4). After 1 h, the patties were grilled in the same way as the chicken breasts (see section above). 

### 2.4. Determination of Principal Components 

The principal components as meat quality parameters were analyzed in the offcuts of cut chicken breasts. The offcuts of each chicken were combined and used for the determination of dry matter, minerals, protein, and hydroxyproline according to the *Amtliche Sammlung von Untersuchungsverfahren* [20] (L 06.00-3 for dry matter/moisture, L 06.00-4 for minerals (ashes), L 06.00-6 for fat, L 06.00-7 for protein, L 06.00-8 for 4-hydroxy proline), which are in accordance with the AOAC. Official Methods of Analysis, [21]. All analyses were performed in duplicate. 

### 2.5. Determination of Heterocyclic Amines

The method includes the determination of 15 HAs. The modified method of HPLC analysis used in the present study [22,23] is based on the method provided by Gross and Grueter [21]. A mixture of IQ, IQx, MeIQ, MeIQx, 4,8-DiMeIQx, 7,8-DiMeIQx, was made in methanol (each component in the range 0.13–0.25 ng/µL). A second mixture of the fluorescence active HAs Glu-P-2, Glu-P-1, norharman, harman, Trp-P-2, PhIP, Trp-P-1, AαC, and MeAαC was also made in methanol (each in the range 0.05–0.1 ng/µL) These solutions were mixed together, and 100 µL of this mixture was used to spike the chicken samples. A caffeine solution (100 µL of a 2.5 µg/mL solution in ultrapure water–methanol, 1:1, *v*/*v*) was added as an internal standard. A total of 50 g of 1 mol/L NaOH was added to approximately 20 g of each finely mixed chicken samples. The mixture was then homogenized using an Ultra Turrax T-25 (IKA Labortechnik, Staufen, Germany) (2 min, 24,000 rpm). To 10 g of each alkaline homogenate, 2.75 g of diatomaceous earth was added and mixed. Further determination was done as previously described [22,23].

The HPLC equipment used for this analysis was a Gynkotek HPLC system (Gynkotek, Germering, Germany) Pump M480, autosampler Gina 50, degasser (DG 1310 S), equipped with a fluorescence detector (RF 1002), diode array detector (UVD 320), and the Gynkosoft chromatography data system (version 5.50). The HAs were determined using the guard column Supelguard™ LC-18-DB (Supelco, Bellefonte, PA, USA) and the TSK-gel^®^ ODS-80™ column (reversed-phase C18, 4.6 mm Id, 250 mm, 5 µm; Tosoh Bioscience, Stuttgart, Germany). The mobile phase consisted of 3 eluents: (A) 10 mM triethylamine phosphate buffer (pH 3.2), (B) 10 mM triethylamine phosphate buffer (pH 3.6), and (C) acetonitrile. The gradient program and UV and fluorescence detection were previously described [22,23]. An injection volume of 40 µL for the non-polar fraction, and 80 µL for the polar fraction, was used. The peaks of the HAs, as well as of the norharman and harman β-carbolines were identified by comparing the retention times and UV spectra with standards. The quantification was carried out with the method of standard addition and for β-carbolines with an external calibration [22,23]. 

### 2.6. Determination of Creatine, Creatinine, and Glucose

Creatine, creatinine, and glucose were determined enzymatically using test kits and following the instructions of the manufacturer (Roche diagnostics GmbH, Mannheim, Germany). The glucose (L 07.00-22) was analyzed according to the instructions of the *Amtliche Sammlung von Untersuchungsverfahren* [20]. For the analysis of creatine and creatinine, 20 g of each minced chicken breast or patty and 100 mL double distilled water were homogenized with the Ultra Turrax T-25 (IKA Labortechnik, Staufen, Germany) for 2 min (24,000 rpm) and diluted to 200 mL with water. The mixture was kept in the refrigerator for 20 min at 18 °C and then filtered through a folded filter (Macherey-Nagel, Dueren, Germany). To precipitate the proteins, perchloric acid (1 mol/L) was added in the ratio of 1:1 (*v*/*v*) to the filtrate. The filtrate was neutralized to pH 6.5 with KOH (6 mol/L), and the clear solution was used for the determination [24].

### 2.7. Color Measurements and Visual Sensory Test

Color measurements of the grilled chicken patties (n = 6) of each treatment were performed after 1 h of the grilling process and then periodically at intervals of 2 min. The L*, a* and b* values in the CIE tristimulus color space were measured using a Minolta colorimeter (Konica Minolta, Langenhagen, Germany) with a standard illuminant D65 in the range of 400 to 700 nm. A sensory visual test was carried out directly after the grilling of the patties (4 patties for each concentration). Trained testers (*n* = 8) evaluated the color of the grilled chicken patties by taking a “like to eat” or “do not like to eat” decision (*n* = 4 for each treatment) following a modified in-out-test as previously described by Munoz et al. [25]. There, the sensory panel had to assess whether a product was within (in—accepted) or out of the norm (out—rejected). In the present study, the panelists had to rank the samples by color according to the 7 different coating compositions (increasing glucose/ribose concentrations have a strong influence on the browning reaction during grilling) and highlight the preferred samples. The test panelists were trained prior to each grilling session for the color assessment using three different colored patties. 

### 2.8. Statistical Analysis

Duplicate samples were analyzed, and measurements were repeated at least twice. Means and standard deviations were calculated using Excel (Microsoft, Redmond, WA, USA). Further statistical analyses were conducted using one-way ANOVA by Sigma Plot 12.5 (Systat Software Inc., San Jose, CA, USA). As an assumption, the data of the product properties were tested for normality and equality of variances (for rejects *p* < 0.05). The samples, which were not normally distributed and/or of unequal variance, were tested using a non-parametric Kruskal-Wallis analysis of variance on ranks using the Student–Newman–Keuls or Dunn’s test. Differences were considered as significant for *p* < 0.05.

## 3. Results and Discussions

### 3.1. Determination of Proximate Composition of the Raw Material

The various uncooked chicken breasts used in the present investigations showed only minor differences in mineral (ash), protein, fat, and moisture content (Table 1), and concentrations were found to be similar to those reported in other studies [26]. 

Connective tissue protein, which is determined through hydroxyproline analysis, contributes to meat tenderness and hence, to the quality attributes of meat [27]. It was the only parameter with a slightly higher variation coefficient of 9.8%. However, the reproducibility of the analytical method was proven in an inter-laboratory ring test with a value of 0.112 g/100 g for connective tissue protein (0.014 g/100 g for 4-hydroxyproline) [20]. 

### 3.2. Occurrence of Heterocyclic Amines in Chicken Breasts

Although 15 HAs were analyzed, only the HAs MeIQx, PhIP, harman, and norharman could be detected. While the latter three HAs were found in all the grilled chicken breasts, MelQx could only be detected in two of the samples (up to 1.12 ng/g). The noteworthy concentrations of PhIP ranged from 1.49 to 9.1 ng/g (Figure 1A). Norharman and harman both had similar levels of between 0.74 and 2.3 ng/g and 0.55 and 2.2 ng/g, respectively (Figure 1B). Both β-carbolines have been found not only in meat but also in plant-based products, including used coffee grounds and edible and medicinal plants, with the highest concentrations detected in tobacco smoke [28]. During alkaloid biosynthesis, harman is easily generated from tryptophan and precursors such as pyruvate or acetate in plants [28]. 

The grilled or fried chicken breasts were found to have particularly high PhIP concentrations (Figure 1A), which is in line with results reported in other studies [14,26]. In comparison to pork or beef containing 4 to 16 times higher concentrations of glucose, chicken has lower concentrations of glucose [13,29], which explains the inhibition of PhIP formation, since high concentrations of glucose (molar ratio of total creatine and glucose >0.5) were found to reduce PhIP formation and, hence, its content in model systems [30]. In Table 2, the concentrations of the detected HAs in grilled chicken breasts, and the corresponding relative variation coefficients are shown. Generally, the relative variation coefficients varied highly, with values of about 84%, 60%, and 30% for MeIQx, PhIP, and both β-carbolines, respectively. In a previous study, the concentrations of PhIP and MeIQx found in chicken meat patties that were prepared from 36 chicken breasts varied only slightly and had relatively low variation coefficients of 14% and 21%, respectively [13]. 

### 3.3. Influence of Precursors on Formation of Heterocyclic Amines

The concentrations of glucose determined in raw chicken breasts was on average 0.35 ± 0.16 g/kg (Table 1) (the range was 0.09 and 0.62 g/kg), In a recent study, raw chicken breasts were found to have generally very low concentrations of glucose (0.18 ± 0.1 g/kg) compared to other types of meat, including horse meat (3.75 ± 0.26 g/kg), beef (1.5 ± 0.77 g/kg) or pork (1.3 ± 0.73 g/kg) [13]. In another study [26], glucose concentrations of 0.4 ± 0.18 g/kg (CV = 43.8%) for 24 chicken breasts were reported, which is in accordance with the present study. However, the total creatine content of the raw chicken breasts varied slightly (CV = 8%). The formation of HAs is influenced by the precursor concentration and the molar ratio between creatine and glucose since both are precursors affecting the formation of HAs [30,31]. A maximal effect on the formation of HAs in model systems could be seen at a molar ratio of total creatine and glucose close to 0.5, with an increasing molar content of glucose leading to a decrease in HA formation [30,31]. In the present study, however, the creatine concentration determined in the chicken breasts was always higher than the glucose concentration, for example, the molecular ratio of creatine/glucose in the 21 samples ranged between 9.6 and 64. The concentrations of creatine were similar in all chicken samples (0.40 up to 0.57 g/100 g) and are in the typical range of the muscle tissue of vertebrates [3]. Due to the slight variations in the creatine content, no significant linear correlation could be found between creatine and any of the detected HAs (*r* = 0.30—PhIP vs. total creatine, *r* = 0.14—MeIQx vs. total creatine, *r* = 0.46—norharman vs. total creatine, *r* = 0.29—harman vs. total creatine, all HAs *p* > 0.05).

Additionally, in comparison to the levels of total creatine, a large variation coefficient of 45% was determined for the glucose concentration in various chicken breasts. For the formation of PhIP, it was shown that phenylacetaldehyde and the aldol condensation product of phenylacetaldehyde and creatinine are key intermediates in dry heated model systems [15]. In a water-based model, creatinine, glucose, amino acids, and carnosine reacted to PhIP in a first-order-like process [32]. However, the reactions of HAs are not yet completely known, because other food compounds such as antioxidants or pro-oxidants, including oxidized fats and reactive carbonyls, additionally influence the reactions [33].

The only significant linear correlation existed between PhIP and the molar ratio of creatine/glucose (*r* = 0.88, *p* < 0.001) (Figure 2A), with increasing glucose contents generally reducing the detectable concentrations of PhIP (PhIP vs. glucose, *r* = 0.69, *p* < 0.001). This result agrees with a previously performed study that focused on different levels of glucose in ground patties of various animal species (*r* = 0.87, *p* < 0.001) [13]. In contrast to this study, the concentration of MeIQx reported here did not significantly correlate with the ratio of creatine/glucose (*r* = 0.18, *p* > 0.05). Moreover, the formation of β-carbolines did not significantly correlate with the glucose concentration. In contrast to PhIP- and MeIQx-formation, the formation of the harman and norharman (β-carbolines) does not need creatine as a precursor [3,28]. β-Carbolines are neither mutagenic nor carcinogenic; however, they may enhance the mutagenic activity of other HAs or other aromatic amines and are therefore considered co-mutagenic [34]. Furthermore, they can be formed at lower temperatures and have been already detected in cooked meat, fish, and meat extract [35].

It has been shown that tryptophan and glucose are precursors, with glucose clearly enhancing the formation of harman and norharman [28]. In Figure 2B, harman and norharman concentrations are presented as a function of different glucose concentrations of 0.7 to 2.3 and 0.6 to 2.2 mmol/kg, respectively. No clear correlations between glucose concentration and harman (*r* = 0.26, *p* > 0.05) or norharman formation (*r* = 0.07, *p* > 0.05) was found in the chicken breasts analyzed. The concentrations of norharman and harman detected in the present study are similar to the ones reported by Pfau and Skog in pan-fried chicken breasts (190–220 °C for 12 up to 18 min) with 1.9–3.5 ng/g and 1–5.7 ng/g for norharman and harman, respectively [28]. Other authors have reported on lower concentrations of norharman (0.01 ng/g) and harman (0.07 ng/g) when using a griddle plate grill [26]. 

### 3.4. Influence of Coating on Formation of Heterocyclic Amines in Chicken Patties 

For the confirmation of our hypothesis that a coating of chicken breast patties with glucose or ribose-containing emulsions results in a reduction of PhIP formation during grilling, a mixture of chicken breasts was mixed together and ground to prepare patties with the same size and weight. Before grilling, the patties were coated with a W/O emulsion containing different levels of ribose or glucose. 

The results of the coating experiment are shown in Table 3. Coating the chicken patties with emulsions containing 1% to 20% glucose resulted in an inhibition of PhIP formation by approximately 22% to 46% (*p* < 0.05), while ribose showed no clear influence on PhIP concentration. Only W/O emulsions with a content of 2.5% and 10% ribose showed an inhibitory effect on the formation of PhIP and a significant difference to the control (*p* < 0.05). One reason might be the high standard deviation of PhIP in the control sample. However, there is a highly significant linear correlation between amounts of ribose and glucose and the rising concentrations of norharman (*r* = 0.92, *p* < 0.01 and *r* = 0.98, *p* < 0.001, respectively) and harman (*r* = 0.98, *p* < 0.001 and *r* = −0.72, *p* < 0.05, respectively). Moreover, no significant linear correlation was found between ribose and PhIP (*r* = −0.37); but there was one between glucose and PhIP formation (*r* = −0.56, *p* < 0.01). After grilling, the concentration of the precursor creatinine increased to 44 ± 12 mg/100 g in the controls and also in patties that had been coated with W/O emulsions containing either ribose or glucose. No significant differences regarding the levels of creatinine could be found between the control and the coated patties after heating (*p* > 0.05). 

These sugars are the most common saccharides in chicken meat [36,37]. ATP degradation products in chicken begin to accumulate in the post-slaughter phase and during refrigerated storage. In particular, ribose and ribose-containing metabolites, such as inosine, inosine-5′monophosphate, and hypoxanthine, are formed [36]. However, glucose and glucose-phosphate are known to degrade in the post-mortem glycolysis, which is why only low concentrations of both components can be found after the common chilled storage performed after slaughtering [36]. Consequently, chicken contains very low concentrations of glucose in comparison to meats of other species [13]. In chicken breasts, the sugars glucose and ribose as well as glucose phosphates and ribose phosphates were found with the respective concentrations of 404 ± 177 mg/kg (CV = 43.8%), 247 ± 84 mg/kg (CV= 33.9%), 137 ± 63 mg/kg (CV = 49.5%), and 169 ± 135 mg/kg (CV = 79.7%) [37]. Furthermore, ribose and its metabolites are important flavor compounds for chicken, which are present in higher amounts than in meats of other species [36]. Coating with emulsions (marinades) is carried out for a variety of reasons, including the improvement of tenderness, flavor, and juiciness of the cooked meat products [38].

In some model systems, in which an aqueous mixture had been treated at low temperatures for a longer period of time (37 °C and 60 °C for 1–2 weeks), PhIP was generated even when a mixture of L-phenylalanine, creatinine, and D-ribose or D-glucose was present in the samples [39]. In these experiments, the sugars were necessary for the formation of PhIP. The influence of four different concentrations (0.15–1.35 mmol) of glucose, fructose, sucrose, and lactose on PhIP formation in model systems have been previously reported [40]. In this study, the addition of glucose, fructose, sucrose, and lactose resulted in inhibitions of 26.9–97.1%, 50.0–96.0%, 20.5–70.6%, and 47.3–97.6%, respectively. Glucose and fructose (1.35 mmol) and the disaccharide lactose (1.8 mmol), all reducing sugars, showed equally high inhibitory effects on PhIP formation in these model systems, which contained twice the molar concentration of creatinine or phenylalanine (0.9 mmol) [40]. In another study, Shin et al. observed that the addition of fructose or glucose resulted in an inhibitory effect of PhIP formation (42.2% and 41.1%, respectively). However, sucrose as a non-reducing sugar did not cause a significant inhibition of PhIP formation in ground beef patties (225 °C for 10 min/side) [41]. Additionally, these authors showed that a combination of fructose and glucose in three different types of honey did not result in a synergistic effect on inhibition [41]. 

The effect of ribose on HA formation has been investigated very rarely. In model studies, MeIQx was formed by using a mixture of creatine, ribose, and alanine or lysine. α-Dicarbonyl compounds are mainly formed from monosaccharides and disaccharides in food by enolization and dehydration reactions and have shown an inhibitory effect on PhIP formation [42]. In meat extract model systems, the reaction with α-dicarbonyl compounds such as methylglyoxal showed that methylglyoxal clearly enhanced the formation of norharman and harman, as well as slightly enhancing that of MeIQx [43]. Additionally, different types of sugars (table sugar, brown sugar, and honey) in a marinating formulation were shown to have an inhibitory effect on the formation of PhIP [38]. Glucose was identified as an enhancer of the formation of norharman, and tryptophan was shown to be a precursor of norharman [35]. Increasing the contents of reducing sugars and the relatively constant concentration of creatine in the muscle tissue might change the formation pathway of PhIP during the Milliard reaction, which might then result in a larger number of intermediates and other reaction compounds, but also in lower amounts of PhIP [44]. The hypothesized inhibitory effect of glucose and, partly, of ribose on the formation of PhIP was confirmed. On the other hand, the formation of β-carbolines was partly enhanced by adding the sugar emulsions to the surface of the chicken patties. 

### 3.5. Color Measurement and Visual Sensory Perception of Marinated and Grilled Chicken Patties 

In Figure 3 and Figure 4, the color change of the grilled chicken breasts containing increasing glucose and ribose concentrations is presented. The color of the crust changed from a more pleasant reddish-brown to a black surface with increasing concentrations of saccharides. In particular, the L*-value changed significantly with increasing concentrations of ribose and glucose (*p* < 0.05) to lower L*-values (Figure 4A). 

The a*-values of patties coated with emulsions containing 20% ribose were similar to the control. Additionally, the a*-values of patties coated with 1–2.5% glucose containing emulsions had similar values (Figure 4B). The b*-values significantly decreased to approximately 5 and 10 with ribose and glucose, respectively (Figure 4C). During the Maillard reaction, reducing sugars and amino acids react to give glycosylamines. After the Amadori rearrangement, ketoamines, and after further dehydration and deamination, dicarbonyls, are formed. Dicarbonyls react further with intermediate compounds to brown polymers (melanoidins) [44]. In a visual sensory evaluation, a group of eight trained panelists evaluated 28 grilled patties using the in and out method [25]. They rejected more than 95% of the patties coated with 10% and 20% saccharides in W/O emulsion. All testers ranked four times the patties in the order from very light to very dark according to the increasing concentration of the saccharides. They preferred chicken patties that had been coated with emulsions containing <2.5% saccharides, with a clear preference for coated patties containing either 0.5–1% ribose or 1% glucose in the W/O emulsion. The preferred saccharide levels led to a reduction of PhIP by 28–34% for ribose and by 39% for glucose, respectively.

The use of both glucose and ribose in W/O emulsion can help to reduce the PhIP concentration, and may hence improve the color on the surface of chicken. Because higher concentrations resulted in a darkening and blackening of the product surfaces, the practical application of both sugars is limited to the use of emulsions with a maximum concentration of 2.5%. 

Due to these afore-mentioned variations, a predictive assessment of human intake of HAs after consumption of a specific amount of grilled chicken is difficult. An accurate risk assessment of the daily exposure of these harmful substances has to focus on the amounts of precursors in the raw material and consider the combination of preparation and cooking applications. 

## 4. Conclusions

The IARC has recommended minimizing the daily intake of HAs, which are found in cooked meat and meat products. The natural variation of HA precursors in chicken breasts may explain the widespread range of HA concentrations, such as PhIP in grilled chicken breasts, even when the preparation procedures (time, temperature) are kept constant. The high variation of glucose clearly influenced the formation, and hence the concentration of PhIP. The inhibitory effect of reducing sugars on PhIP formation was confirmed by coating the chicken patties with W/O emulsions containing increasing concentrations of glucose or ribose. In the formation mechanism of PhIP, the molar ratio of total creatine and glucose was shown to play a key role, while this was not the case for β-carbolines. In conclusion, the presented results of HA contents in chicken breasts, in combination with dietary assessments, may allow for a better estimation of HA exposure when (grilled) chicken breasts are consumed.

## Figures and Tables

**Figure 1 foods-08-00616-f001:**
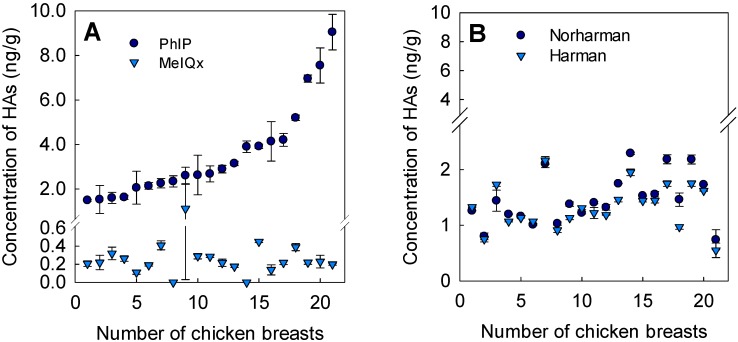
Concentrations of heterocyclic amines (Has) in 21 different grilled chicken breasts sorted in ascending order of the PhIP levels. (**A**) Concentration of PhIP and MeIQx; (**B**) Concentrations of β-carbolines norharman and harman.

**Figure 2 foods-08-00616-f002:**
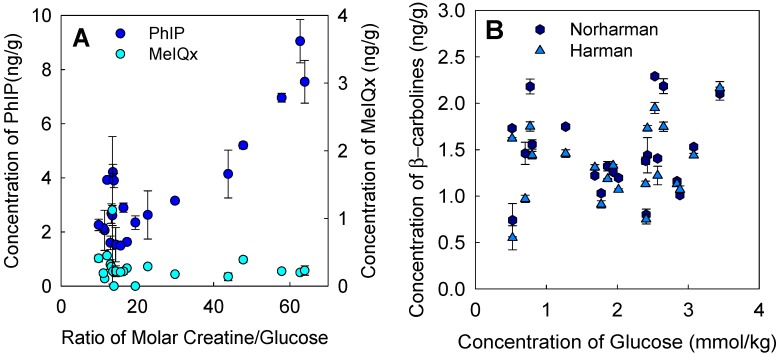
(**A**) Concentrations of PhIP and MeIQx of different chicken breasts as a function of the ratio of creatine/glucose. (**B**) Concentrations of norharman and harman as a function of the glucose content.

**Figure 3 foods-08-00616-f003:**
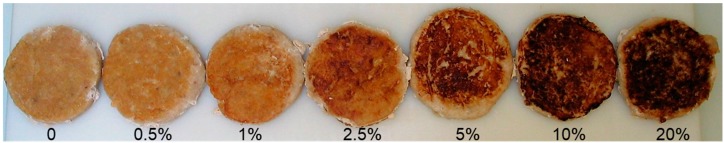
Image of emulsion-coated and grilled chicken patties using different glucose concentrations (0%, 0.5%, 1%, 2.5%, 5%, 10%, and 20%, *w*/*w* emulsions).

**Figure 4 foods-08-00616-f004:**
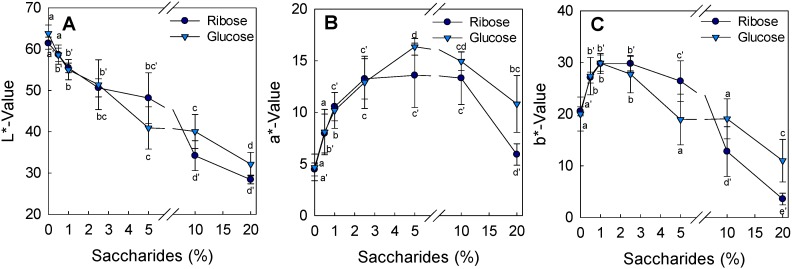
Color measurement of grilled chicken patties after coating with saccharides in different concentrations (**A**) L* value (lightness), (**B**) a* value (red), (**C**) b* value (yellow). Means (a’–e’ ribose, a–e glucose) with different letters are significantly different (*p* < 0.05).

**Table 1 foods-08-00616-t001:** Chemical composition of the raw chicken breasts before grilling.

Analysis	Mean ± SD ^a^ (g/100 g)	CV ^b^ (%)
Protein	23.9 ± 0.25	1.0
Moisture	74.2 ± 0.58	0.8
Lipids	1.57 ± 0.12	7.6
Minerals (ash)	1.18 ± 0.04	3.3
Connective tissue protein	0.51 ± 0.05	9.8
Glucose	0.035 ± 0.016	44.6
Total creatine	0.45 ± 0.039	8.6
Creatinine	0.021 ± 0.01	4.8

^a^ SD: standard deviation, ^b^ CV: relative variation coefficient.

**Table 2 foods-08-00616-t002:** Concentrations of HAs in grilled chicken breasts (*n* = 21).

HAs	Minimal Level (ng/g)	Maximal Level (ng/g)	Mean ± SD ^a^ (ng/g)	CV ^b^ (%)
PhIP	1.49	9.05	3.52 ± 2.10	59.5
MeIQx	n.d. ^c^	1.12	0.27 ± 0.23	83.3
Norharman	0.74	2.29	1.46 ± 0.44	30.3
Harman	0.55	2.16	1.33 ± 0.40	30.1

^a^ SD: standard deviation, ^b^ CV: relative variation coefficient, ^c^ n.d.: not detected (<0.05 ng/g).

**Table 3 foods-08-00616-t003:** Concentrations of HAs (means ± standard deviation) in grilled chicken patties after coating with various amounts of saccharides in water in oil (W/O) emulsions.

Saccharide	Concentration (%)	PhIP *^a^* (ng/g)	MeIQx (ng/g) *^b^*	Norharman *^a^* (ng/g)	Harman *^a^* (ng/g)
Ribose	0	1.74 ± 0.42 ^a^	n.d.	0.22 ± 0.01 ^a^	0.10 ± 0.01 ^a^
	0.5	1.15 ± 0.11 ^a,b^	n.d.	0.22 ± 0.01 ^a^	0.10 ± 0.01 ^a^
	1	1.26 ± 0.31 ^a,b^	n.d.	0.23 ± 0.01 ^a^	0.39 ± 0.01 ^b^
	2.5	1.07 ± 0.25 ^b^	n.d.	0.32 ± 0.03 ^b^	2.96 ± 0.03 ^c^
	5	1.44 ± 0.35 ^a,b^	n.d.	0.39 ± 0.02 ^c^	2.52 ± 0.01 ^d^
	10	1.06 ± 0.12 ^b^	n.d.	0.72 ± 0.03 ^d^	5.48 ± 0.02 ^e^
	20	1.19 ± 0.08 ^a,b^	n.d.	4.81 ± 0.89 ^e^	16.76 ± 0.46 ^f^
Glucose	0	1.58 ± 0.13 ^a^	n.d.	0.30 ± 0.01 ^a^	0.69 ± 0.01 ^a^
	0.5	1.55 ± 0.17 ^a^	n.d.	0.22 ± 0.01 ^b^	1.00 ± 0.20 ^b^
	1	0.97 ± 0.06 ^b^	n.d.	0.30 ± 0.01 ^a^	1.04 ± 0.05 ^b^
	2.5	0.89 ± 0.22 ^b^	0.20 ± 0.01	0.81 ± 0.03 ^c^	0.66 ± 0.01 ^a^
	5	1.24 ± 0.12 ^c^	0.22 ± 0.04	0.81 ± 0.03 ^c^	0.17 ± 0.01 ^c^
	10	0.89 ± 0.06 ^b^	n.d.	1.33 ± 0.02 ^d^	0.20 ± 0.03 ^c,d^
	20	0.86 ± 0.05 ^b^	n.d.	2.06 ± 0.01 ^e^	0.22 ± 0.01 ^d^

*^a^* Means with different letters (a, b, c, d, e and f) are significantly different (*p* < 0.05) *^b^* n.d.—not detected.

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
