# Peer review of "Effect of Creatine and Glucose on Formation of Heterocyclic Amines in Grilled Chicken Breasts"

_foods, 2019, doi:10.3390/foods8120616_

Round 1

Reviewer 1 Report

Despite the obvious bias of not providing the context that all foods containing amines, whether from plants or animals, might develop heterocyclic amines from cooking, this experiment is severely flawed from the sampling and sample preparation perspectives. Additional information is needed on the source of the chicken breasts as breed, age, and even sex of the bird will influence composition, which will then change the compounds derived from heating. There was no justification for use of safflower oil and aluminum foil for the cooking of samples, which could impart impurities contributing to the HA levels. The homogenization of the entire breast for sampling rather than surface and interior sampling provides confounding of the results of the cooking process. There was no control, i.e. without treatment with oil, foil, or marinade, for either the breast or patty experiments.

There was some missing information in the materials and methods that would prevent duplication by other scientists. The results and discussion section lacked scientific explanations of the results and discussion of the results compared with those of other studies..

Line(s)           Comment

 32-33           Reference is needed to document this statement as factual.

33-48           It was expected that the minimum concentration to initiate carcinogenesis or mutagenesis from heterocyclic amine and similar compounds would be provided as a reference point for the relative danger of consuming these compounds.

 34-37           Reference(s) is needed to document these statements as factual.

 59-60           “a hypothesis”

  82                More details are needed on the source of the chicken breasts as bird age will influence composition and thus change the components derived after heating.

 86-87           Frying and grilling are two different heating processes so consistency in terminology is needed.

94                 The section is titled “preparation and marinating (sic) of the chicken patties”, but this indicates that the patties were coated with an emulsion. The authors should refer to coating the patties with the sugar, water, and oil emulsion as marinating means soaking or steeping in a solution.

102-106        Evidence must be presented that the composition of the residue from the cutting of cubes from the breasts is representative of the breast cube (and patty) composition since composition will vary anterior to posterior, proximal to medial, and dorsal to ventral portions of the Pectorales major muscle.

106                A reference in English or a more detailed description of the methods for proximate analysis must be provided.

149                The number of testers to determine preference or acceptability of color is very lacking as 50 is the minimum acceptable number of panelists for consumer tests. Use of either trained or consumer panel sensory testing requires details of the use of human subjects, including the approval for use of humans in research; the process of selecting the testers; the number of training sessions, samples used for training, the number of samples in each training session for trained panels; the number of testing sessions; number of samples in each testing session; the randomization of sample identification; and the environmental conditions of room temperature, lighting, time of day.  

162-164 and Table 1 It is incorrect to state that this composition is of the raw chicken breasts since the residue and not the cubes or samples of patties were analyzed.

165-166        Reference is needed to substantiate the validity of this statement.

166-168        A scientific explanation for the variation and the potential influences on the results that were observed should be proposed.

177                The results are in ppb so a context of the relative safety risk from these levels and the relative amounts of Has produced by cooking plant foods containing amines must be provided.

180                “formation with high levels of glucose”?

Figure 1        An explanation should be provided for why 20 chicken breasts would have 4.5 times higher PhIP levels than 5 chicken breasts rather than only 4 times higher (2 x 4 = 8; 2 x 4.5 = 9).

186                Additional explanation is needed for the differences in samples, and hence the results, between this study and the Gibis and Weiss 2015 study.

191                The glucose contents of the other types of meat are not shown in table 1 and so must be presented here or the sentence becomes speculative rather than definitive.

217                A scientific explanation for the difference in the current results compared with those of Pfau and Skog 2004 on Harman and Norharman levels should be provided.

223-224        This sentence must be reworded since the patties were coated and not marinated and the emulsion was not a marinade.

225-229        This explanation is not valid unless the time postmortem of the chicken used in the study is given.

238                A scientific explanation should be provided for the inhibition of PhIP formation with 2.5 and 10 % ribose, but not with levels higher or lower than these and why 5 % level of glucose did not inhibit PhIP levels while other levels of glucose higher than 1% caused inhibition.

259-261        Scientific explanations for the difference in the two reducing sugars on the formation of PhIP and the enhancement of carbolines by sugar addition must be provided.

263-271        The scientific explanation for heating of sugars and amines to cause Maillard browning is expected.

271-272        Because there were only 8 testers to determine like and preference, this sentence must be deleted.

279                It is redundant to have “significantly” and the probability level in the same sentence since the probability level has already determined the level of significance.

284-285        Because the sensory results are invalid due to the limited number of testers, it cannot be stated that the color was improved.

289-291       This sentence is sufficiently vague and has no meaning, particularly since chicken is not eaten raw.

293                The study did not analyze the HA precursors of chicken breasts nor were there correlations between any proximate analysis values and HAs in each chicken breast presented in the paper.

295-298        These sentences reiterate results and do not provide conclusions based upon the results.

Author Response

We thank the reviewer for his/her assessment. The comments below were fully considered.

Reviewer 2 Report

I have only some minor editorial remarks:

Introduction: As the IARC classified red meats rather than white meats as probably carcinogenic, a word of explanation why the authors studied white meat would be useful.

Lines 48, 55, 263-264: some linguistic correction is recommended ("chicken using the same preparation method", "dissimilar to the quantities", "the color alteration is shown that the color changed...".

Line 85: do you really mean "tin foil"?

Line 244: I suggest to put "37°C and 60°C" in brackets ()

Author Response

We thank the reviewer for his/her assessment of the presented work. His/her comments below were fully considered.

Reviewer 3 Report

Thanks for your efforts.

The effect of inherent creatine and glucose contents on the formation of HAs after grilling of chicken breast. The ways to reduce carcinogenic compounds in meat are very important, and this manuscript has good information. However, the results of the effect of ribose and glucose on the formation of PhIP, Harman, and Norharman were not clearly discussed. In addition, some sentences had grammatical errors.

I attached my opinions on your manuscript.

L10: Please delete “ by using an electric contact plate grill” I think it is just cooking equipment.

L117: Collect to “The 50 mL of 1 M NaOH”

L153: How many times did you repeat the experiment for marinated chicken? If you repeat two times, it is not enough.

L190: Why was there no data of creatine effect on the HAs formation? Please suggest an independent effect of creatine and glucose contents on the formation of each HA.

L214-217: Author found the different effect of glucose on Norharman and Harman formation compared to that from another study. Please suggest the reasons for it

L221: The margination levels of glucose and ribose influenced the Norharman and Harman formations. However, the result was not explained, and the discussion was not suggested in this part. In addition, no effect of glucose on the Norhrman and Harman formations was found in Figure 2. However, Table 3 showed different results.

L281-283: I could not understand this explanation. Why was it different with L271-272

292: Please suggest a clear conclusion. The content of this part was the re-explanation of results.

Author Response

First we would like to thank the reviewer for his/her assessment.

Round 2

Reviewer 1 Report

It is appreciated that the authors attempted to address reviewer comments in the revised manuscript. However, the scientific flaws are sufficiently severe. The basis of the research was that carcinogenicity or mutagenicity of HAs occurs with ppm levels, but the HA levels measured in the control samples were ppb, or a thousand-fold difference in the levels that would cause health detriments. The authors fail to explain the toxicity (or lack thereof) with the observed levels of HAs. The conclusions are written to justify the research that was conducted rather than being accurate and scientifically correct that the results have no practical meaning despite any statistical differences in the results. The second flaw is that color was determined for like or preference by trained testors. These sensory results as like or preference can only be determined by subjective testing by consumer panels and recognized scientific bodies like the Society of Sensory Professionals, International Institute of Food Science and Technology, Institute of Food Technologists, and ISO clearly state the differences and requirements for affective and descriptive sensory testing.

Author Response

Thank you for your valuable comments and suggestions. We have tried to explain our choice of the sensory test method.  We hope that our responses (attached word file) are appropriate to offer useful information to the journal’s audience.

Reviewer 3 Report

The author answered that the experiment was repeated 3 times. Why was the n-number 4?

Author Response

Thank you for your valuable suggestions, time, and your patience.
